# Exploring stigma experiences of scattered-site public housing residents and its characteristics based on social contact theory

**Sungik Kang**[ID][☯]**, Ja-Hoon Koo**[ID]*[☯]

Department of Urban and Regional Development, Hanyang University, Seoul, South Korea

☯ These authors contributed equally to this work.
* jhkoo@hanyang.ac.kr

**Data Availability Statement:** The data and Stata code used for these analyses can be accessed via the following link: https://figshare.com/account/articles/26962807.

## Abstract

Governments worldwide have been striving to efficiently manage public rental housing. However, the stigma associated with public rental housing persists as a significant challenge. In response, the scattered-site public housing strategy has been introduced as an alternative to traditional large-scale rental housing. The objective of this study was to evaluate the effectiveness of this strategy in reducing the stigma within Seoul metropolitan city. The empirical analysis utilized 2019 Seoul Public Housing Occupant data and a binary logistic regression model. The main findings indicate that residents of scattered-site public housing experience significantly lower levels of stigmatization compared to residents of other public housing types. Notably, the stigmatization experienced by scattered-site public housing residents is lower not only compared to independent public housing residents but also to those in socially mixed public housing, which is typically advantageous for reducing stigmatization. This suggests that residents of scattered-site public housing are statistically more free from both external and internal stigmatization. In addition, a unique characteristic found only in scattered-site public housing is that as residents form closer relationships with their neighbors, they experience more stigmatization. This implies that as scattered-site public housing residents form closer relationships with their neighbors, their identity as public housing residents can become exposed, potentially leading to increased stigmatization.

## Introduction

Countries worldwide are making efforts to stabilize housing for low-income populations through the efficient management of public rental housing. However, public rental housing residents are exposed to social stigma [1], which is one of the biggest obstacles in operating public rental housing [2–4]. Societal perceptions of bias for public rental housing often lead to the stigmatization and discriminatory treatment of the residents. For example, the negative perception associated with public rental housing may be linked to concerns about increased crime and a decline in housing prices [4]. Additionally, other negative perceptions persist, such as the belief that poor people contribute to deteriorating living environments and that neighborhoods become deteriorated due to poverty [3, 5–7].

**Funding:** This work was supported by the Korea National Research Foundation under Grant [RS-2023-00239319]. The funder had a role in study design, data collection, empirical analysis, and drafting the manuscript.

**Competing interests:** The authors have declared that no competing interests exist.

Negative perceptions of public rental housing are related to the quality of the building and the socio-demographic characteristics of the residents [3, 8, 9]. Public rental housing is often characterized by low design quality and high-form construction, which are related to stigma perceptions [4, 10, 11]. This unique form of public rental housing, which differs from general housing, serves as a stigma anchor [3, 4]. In addition, public rental housing is intended to provide stability for marginalized populations, with large-scale public rental housing complexes frequently housing a concentration of low-income residents [5, 6]. Due to the concentration of marginalized populations and associated concerns about deteriorating living conditions, public rental housing complexes frequently face societal stigmatization [4, 12, 13].

Various strategies have been implemented to resolve this stigma. Socially mixed public housing allows public housing residents to live with middle to upper-class homeowners, avoiding stigmatization by not marking the units as public housing [2]. However, internal stigmatization from the homeowners persists [1]. To address this issue, scattered-site public housing was newly introduced. Scattered-site public housing means smaller-scale units dispersed in general residential areas, overcoming the limitations of traditional public housing by adopting a strategy of being small-scale, low-density, and widely scattered [14, 15]. These units blend into the surrounding neighborhood with similar designs, making them indistinguishable as public housing [16]. This helps residents avoid the stigma and social exclusion associated with traditional public housing. Scattered-site public housing also enables low-income residents to access a wider range of social and economic resources, improving their living standards [15]. This form of public housing is provided not only in the United States, Europe, and Australia but also in several Asian cities. According to a study by de Souza Briggs et al. [17], scattered-site public housing increases residents' housing satisfaction, promotes community integration, and does not negatively impact property values in the surrounding area.

Scattered-site public housing is being supplied worldwide, and various studies have been conducted on it. However, there is a lack of research that specifically examines the effects of scattered-site public housing (SSPH) on stigmatization from various perspectives. Therefore, the purpose of this study is to explore whether residents of scattered-site public housing experience less stigmatization compared to residents of other public housing types. Specifically, it is significant to analyze the characteristics of stigmatization by comparing the levels experienced by SSPH residents with those experienced by residents in independent public housing (IPH), where only public housing residents live, and socially mixed public housing (SMPH). To conduct this empirical analysis, the methodology involves first addressing data imbalance and then using binary logistic regression to analyze and compare the levels of stigmatization in SSPH with those in IPH and SMPH.

## Literature review

### Introduction and characteristics of scattered site public housing

The social mix strategy of public rental housing was introduced to reduce the concentration of slums, creating a more cohesive community and providing more opportunities for residents of public rental housing [18]. Therefore, the US, the UK and Western Europe have put efforts into developing mixed-income communities [19, 20]. Some research suggested that the social mix strategy had the function of reducing discrimination, stigma, and social exclusion and fostering inclusion, lower levels of crime, and better neighborhood amenities [21–23]. However, opinions differ about the effectiveness of the social mix strategy. Those against the strategy emphasize that social mix public rental housing works differently from its original purposes, such as being divided into rental housing residents [24], little interaction with other classes [25] and social mix rejection due to tax burden [26].

Students living in poor public housing complexes demand public housing that could alleviate poverty and joblessness [27, 28], and the residents demand public housing that could help them access job-related networks and information, mainly in the American society [29]. These requests have led to new discussions about desegregation and affordable housing and provided an opportunity for several local scattered-site public housing programs to operate [17]. Consequently, public housing authorities supplied SSPH by purchasing existing houses and renovating them or constructing cluster-type houses and new apartments [16]. The SSPH has been distributed across communities, with a minimum of 2 to 300 single-family, duplex and triplex homes or garden apartments per site, rather than being concentrated in areas of poverty and segregation [16, 30]. For example, as a dispersed housing strategy, the Denver Housing Authority purchased existing single-family and duplex units in 1969 and rehabilitated them to provide public housing [15].

Unlike the main concerns of existing subsidized housing, such as the decline in housing prices in the surrounding area and the outflow of middle-income classes, Yonkers' SSPH is believed to provide housing with little discrimination against minority communities [17]. In a study targeting the Lakeview community SSPH in Chicago, the residents of the SSPH generated revenue through increased accessibility to job search programs and improved their sense of community [30]. In an SSPH study exploring the relationship with crime, no evidence was found regarding increased crime rate in the area surrounding the SSPH; however the total number of crimes decreased was more significant [15]. However, a study investigating residence satisfaction in the SSPH found that in areas with a high public housing ratio, the difference in residence satisfaction was not large. Thus, in these areas, it was advantageous for the government to cross-use existing public housing and the SSPH [16].

## Stigmatization in public housing and socio-regional contexts

Stigmatization is the process by which a social group categorizes an individual or group with negative stereotypes, resulting in discrimination and exclusion [31]. Social stigmatization refers to negative perceptions and discrimination against specific groups. Public housing and its residents are prime examples of this stigmatization. Public rental housing and social housing complexes are often stigmatized as being for low-income, most disadvantaged, and marginalized people, leading to social prejudice [3]. This negative perception of public housing is reinforced, causing residents to experience negative impacts in their living and social environments [32]. Such stigmatization is associated with the visible features of low-quality building design and high-density structures typical of public housing [33].

According to previous studies, SSPH has an effect on mitigating such stigmatization. Traditional public housing experienced stigmatization due to the collective residence of low-income or minority groups in one place. In contrast, SSPH disperses public housing, allowing low-income residents to live in various areas, which helps prevent negative perceptions of specific regions. Varady and Preiser [16] emphasized that SSPH is designed to integrate the housing form with the surrounding community's architecture, helping to avoid the stigmatization associated with public housing. Santiago et al. [15] demonstrated that the public housing dispersion increases the social capital of low-income residents and promotes interaction with the community. De Souza Briggs et al. [17] found that in SSPH in Yonkers, New York, there were no signs of racial discrimination against public housing residents by nearby residents. Therefore, SSPH has advantages in mitigating the stigmatization of public rental housing, allowing public housing residents to be freer from stigmatization related to their housing status.

Interaction with neighbors and the class structure of socially mixed communities are fundamentally related to the stigma associated with public housing. Social Contact Theory

emphasizes that frequent and positive interactions with stigmatized groups may reduce stigma [34]. This theory posits that contact between in-groups and out-groups (stigmatized groups) can improve the former's attitudes by providing experiences that counteract ignorance and stereotypes [35]. Echoing this, Raynor et al. [2] proposed that individuals who frequently interact with public housing residents and live in socially mixed communities are likely to experience less stigma toward these residents. The size of the out-group also directly influences the attitudes of the in-group, related to the density and frequency of contact [35]. Research by Luu et al. [36] revealed that while residents of market-rate housing accept public housing residents up to a certain threshold, they reject them beyond that point, indicating that the number of public housing households in a neighborhood can contribute to tension and conflict. Furthermore, the economic level of the neighborhood may correlate with discrimination experiences and neighborhood tensions among SMPH residents. Graves [37] found that people generally prefer socioeconomically homogeneous neighborhoods and tend to avoid heterogeneous ones, and this preference can create challenges for SMPH residents in high-income areas in terms of neighbor interactions. Similarly, Sohn and Ahn [38] observed that SMPH residents face more conflict in socioeconomically heterogeneous neighborhoods compared to homogeneous ones.

## Scattered site public housing policy in South Korea

In Seoul, South Korea, the stigma against public rental housing residents is continuously increasing. The Seoul Metropolitan Government began constructing complex-type public rental housing for the poor and low-income classes in 1989, with a significant supply starting in 1991 [39]. In the early stage, public housing supply was concentrated on the outskirts of Seoul, where large-scale housing sites were more readily available [39]. However, this approach led to social problems like increased stigma, social exclusion, segregation, and the formation of slums [40, 41]. Most public rental apartments managed by the Korea Land and Housing Corporation (LH), which is the South Korean government's public rental housing supplier, are part of a complex known as Humansia. The residents of Humansia, often derogatorily referred to as 'Humansia beggars' or 'Elsa' (a shorthand for LH residents), face significant stigma, marking it as a serious social issue [42, 43]. Discriminatory treatment against public rental housing residents, such as restricted access to community facilities or spaces, remains a contentious issue in South Korea, as noted in several studies [25, 44].

To address issues such as stigmatization, concentration of low-income households, and social exclusion arising from large-scale independent public housing, the Korean government has started providing socially mixed public housing, similar to approaches in the United States, Europe, and Australia [45]. Since the late 2000s, the Korean government has been supplying socially mixed public housing districts where public housing residents live alongside market-rate housing residents. However, persistent issues still remain. In SMPH community districts, public housing and market-rate housing residents have not been physically or socially integrated, and the social exclusion of public housing residents by market-rate housing residents continues to be a problem [24]. In response to the issues of stigmatization and social exclusion arising in SMPH, the Korean government has introduced scattered-site public housing in Seoul. This proactive form of social mixing has been implemented through scattered-site public housing, which provides public housing on a smaller scale within existing neighborhoods. The primary purpose of this approach is to improve the quality of rental housing and reduce stigmatization by integrating public housing residents as fully as possible into the general housing community [46]. The supply method is that the LH and Seoul Housing & Communities Corporation (SH) purchase existing single-family or multi-family housing in Seoul and

lease them to low-income individuals, thereby expanding the rental housing supply [39]. By 2018, Seoul had supplied 30,429 units of such public housing, up from 614 in 2002 [39].

## Research contributions and hypotheses

Although various studies have examined the effects of SSPH on resident satisfaction [16], social interaction, and socio-economic status [30], as well as changes in home prices [17], there remains a significant gap in the literature concerning SSPH's role in reducing stigmatization. To our knowledge, no existing studies have statistically compared SSPH with other types of public housing, such as IPH and SMPH, to verify its effectiveness in this regard. By doing so, this study aims to fill this gap by analyzing the characteristics of SSPH related to stigmatization and demonstrating its potential advantages over other housing types. Thus, this study contributes to the existing body of literature by providing a quantitative analysis of whether SSPH offers advantages in reducing stigmatization compared to other types of public housing such as IPH and SMPH. Additionally, this research sheds light on the influence of neighborhood relationships among SSPH residents on stigmatization, an area that has not been extensively explored in previous studies.

The hypotheses of this study are as follows. The first hypothesis is that SSPH actually shows advantages in reducing stigmatization. This analysis distinguishes between IPH and SMPH within public rental housing, as SMPH was introduced as a policy to mitigate stigmatization by mixing higher-income and public rental households within one residential community. However, despite reducing external stigmatization outside the residential district, SMPH also experiences internal stigmatization among residents [1]. Therefore, the second hypothesis is that SSPH shows advantages in reducing stigmatization not only in IPH but also in SMPH. The third hypothesis is that social contact conditions help SSPH residents reduce their experience of stigmatization. According to social contact theory and empirical research, social contact can have positive effects, but excessive contact may also have negative impacts. Hence, the third hypothesis examines the characteristics of stigmatization among SMPH residents based on social contact. The diagnosis of these hypotheses is expected to reveal findings that were not empirically demonstrated in previous studies and may lead to unexpected conclusions.

## Materials and methods

### Seoul public rental housing

In the 1980s, in response to skyrocketing housing rent prices and the widening gap between the rich and poor, the South Korean government initiated the provision of public rental housing to address the housing shortages in Seoul. Initially, amid urgent demand, the focus was on large-scale supply on the outskirts of Seoul [47]. Since 2000, to address challenges associated with large-scale rental housing, the government has implemented a social mix strategy and a decentralized supply approach. Seoul's public rental housing system categorizes tenants based on their socioeconomic levels, offering options ranging from 50-year permanent residences to 10-year leases. Households earning below the first income quintile and low-wage workers are eligible for occupancy (Table 1). According to a report by the Seoul Institute, there were approximately 294 thousand public rental housing units in Seoul as of 2019, accounting for 7.7% of all households in the city [48]. Of these, 206 thousand units are provided by the city government and 87 thousand by the central government. Public rental housing in Seoul is offered by the SH and the LH, with SH-managed units receiving partial funding from central government subsidies. The management of public rental housing is split between SH and LH, with each organization overseeing the housing they supply. The average rent for these units is

**Table 1. Description of Seoul public rental housing programs.**

| Program[a] | Description[a] | Eligibility (income)[b] | Rental period[a] | Supply form |
|---|---|---|---|---|
| Permanent Public Housing | Public rental housing offered at reduced rents to stabilize housing for individuals with incomes in the first income quartile | <50% | 50 year | IPH |
| 50–Year Public Rental Housing | Public rental housing designated for non-homeowners with low incomes | <70% | 50 year | IPH |
| National Public Housing | Public rental housing available to those earning 70% or less of the average worker's wage | <70% | 30 year | IPH |
| Purchased & Leased Public Housing | Public rental housing acquired by purchasing existing homes and then renting them out to low-income individuals | 70–90% | 20 year | SSPH |
| Shift housing | Public rental housing for moderate-income households under the Chonsel lease contract system[c] | 50–70% | 20 year | SMPH |
| Redevelopment Public Housing | Rental housing provided by securing housing units during urban redevelopment projects | 50–70% | 50 year | SMPH |

Source a: 2023 User Guide of Seoul Public Rental Housing Tenant Panel Survey, b: Jeon & Woo(2022) [47]

c: The Chonsel lease contract system is a unique housing rental system in South Korea. Rather than making regular rent payments, tenants place a substantial deposit when moving in, and this deposit is returned in full when they vacate.

IPH = Independent public housing, SSPH = Scatter-site public housing, SMPH = Social mixed public housing

about 31.1% of the prevailing local neighborhood rent, and the government further reduces the housing cost burden by subsidizing bank loan interest rates.

## Data

The data used in this study was derived from the 2019 Seoul Public Housing Occupant Data (SPHD). We utilized the 2019 data because the decrease in sample size was smaller than in the 2021 data, due to the data imbalance purification process and the application of Propensity Score Matching. The SPHD has been annually collected since 2016 by the Seoul Housing & Communities Corporation through surveys, with the results shared with the public to inform the review the public rental housing policies. The sample included 3,000 households were initially part of the 2016 and were subsequently followed up with in the 2019 survey. The SPHD surveyed adult men and women, aged 19 years or older, including heads of household and household members. In 2019, the survey was conducted through one-on-one interviews from September to December. It investigated various aspects such as the awareness of living in public rental housing, socio-economic characteristics, perception of the living environment, community spirit, leisure activities, and health status [49].

The dependent variable of this study included the presence or absence of stigma experienced by individuals in various settings such as neighborhoods, streets, workplaces, and schools. Approximately 7% of the respondents reported experiencing stigma, highlighting a data imbalance between stigmatized individuals and those who did not experience stigma. To mitigate the risk of overfitting due to this imbalance, it was necessary to balance the data [50, 51]. For this purpose, we used the Condensed Nearest Neighbor (CNN) method, which enhances data balance by supplementing random sampling based on machine learning algorithms and considering the characteristics of the sample dataset [52]. According to Wilson [53], the CNN algorithm reduces the number of majority class samples by removing redundant and similar samples. This method ensures that only the most representative samples of the majority class are retained, leading to a more balanced dataset [54]. As a result, the CNN algorithm not only reduces the computational cost but also indirectly mitigates data imbalance by addressing the over-representation of majority class samples. In our dataset, from the 3,158 samples who did not experience stigma, 393 were selected using the CNN method. This

selection adjusted the dependent variable's binomial ratio from 7.3:92.7 to a more balanced 35.4:64.6.

## Measurement

Table 2 presents the dependent and independent variables and their measurements. The dependent variable was stigmatization, which measured whether any of the participants experienced discrimination in the past year due to living in public rental housing. While definitions of stigmatization vary across scholars, the common aspect is that it serves to maintain a hierarchical society by shunning and discrediting individuals who exist outside the norms of mainstream society [31, 55]. Stigmatization refers to inequality in the form of discriminatory treatment of out-groups regarding access to resources, social relationships, and coping behaviours [56, 57]. McCormick et al. [1], who studied the internal stigma associated with socially mixed housing, emphasized that stigmatization based on distinguishing characteristics leads to negative stereotyping, social distancing, assumptions of fundamental differences between groups, and, finally, differential treatment with unequal outcomes. Residents often experience this stigmatization and discrimination associated with living in public housing [1]. Based on this definition of stigmatization, Arthurson [12] and Jun & Han [23] measured stigmatization in public housing as discrimination arising from residence in public housing. Following this

**Table 2. Description of variable statistic.**

| Variable | Mean | S.D. | Min | Max |
|---|---|---|---|---|
| Dependent variables | | | | |
| Discrimination[a] (yes = 1, no = 0) | 0.354 | 0.479 | 0 | 1 |
| Independent variables | | | | |
| Public housing | | | | |
| Scattered site public housing[a] | 0.196 | 0.397 | 0 | 1 |
| Independent public housing[a] | 0.459 | 0.499 | 0 | 1 |
| Social mixed public housing[a] | 0.346 | 0.476 | 0 | 1 |
| Sociodemographic attributes | | | | |
| Age[a] | 57.330 | 16.977 | 21 | 98 |
| Gender[a] (male = 1, female = 0) | 0.406 | 0.491 | 0 | 1 |
| Education[a] (above high school = 1, under = 0) | 0.597 | 0.491 | 0 | 1 |
| Employed[a] (yes = 1, no = 0) | 0.943 | 0.232 | 0 | 1 |
| Self-rate physical health[a] | 3.034 | 1.041 | 1 | 5 |
| Self-rate mental health[a] | 2.131 | 0.635 | 1 | 4 |
| Household condition attributes | | | | |
| Log[a] (household income) | 7.435 | 0.780 | 3.912 | 9.064 |
| Married[a] (yes = 1, no = 0) | 0.387 | 0.487 | 0 | 1 |
| Number of family members[a] | 2.503 | 1.211 | 1 | 7 |
| Disable person[a] (yes = 1, no = 0) | 0.176 | 0.381 | 0 | 1 |
| Living period[a] (year) | 13.147 | 8.034 | 1 | 29 |
| Social mix and relationship attributes | | | | |
| Participate in communities[a] | 0.056 | 0.230 | 0 | 1 |
| Number of close neighbours[a] | 2.707 | 0.611 | 1 | 4 |
| Relationship degree with neighbours[a] | 2.824 | 0.691 | 1 | 4 |
| Regional context attributes | | | | |
| Public housing ratio[b] | 0.509 | 0.500 | 0 | 1 |
| Regional housing price index[b] | 0.589 | 0.492 | 0 | 1 |

approach, this study measured the degree of stigmatization based on the discrimination experience related to residence in public rental housing. The respondents were asked whether they had experienced being ignored or discriminated against due to living in public rental housing, responding with either 'yes' or 'no.' The reported settings for such discrimination included neighborhoods, streets, workplaces, and schools.

As independent variables, SSPH and non-SSPH were the types of public rental housing included in this study. In addition, non-SSPH included IPH and SMPH. We categorized respondents into SSPH, IPH, and SMPH residents based on their responses regarding their type of public housing. We included independent variables based on Raynor et al. [2], who studied stigmatization experiences, and Woo et al. [4], who studied stigmatization awareness. Among the independent variables, socio-demographic variables included age, gender, education level, occupational status and health status. Household condition variables included income, marital status, number of family members, disabled person status and living period.

Additionally, drawing on social contact theory, as illustrated in studies by Lee et al. [35] and Pettigrew [34], which demonstrate how positive neighborhood contact helps to reduce intergroup stereotyping and lower stigma. On the other hand, studies by Lee et al. [35] and Raynor et al. [2] explain that excessive contact with public housing residents can have a negative impact on their acceptance. Based on these prior studies, this analysis model includes variables related to social contact: participation in communities, number of close neighbors, and relationship degree with neighbors. Participation in communities was measured by asking whether respondents had attended any resident meetings within the district in the past year, with yes or no responses. Number of close neighbors was assessed with a question asking how many close neighbors the respondent had, ranging from 1 'none' to 4 'three or more'. Relationship degree with neighbors was evaluated by asking how well the respondent knew their neighbors, with options ranging from 1 'do not know the neighbors well' to 4 'know the neighbors well and frequently visit each other's homes'. For the regional context, we used two variables: the ratio of public housing and the regional economic level. We measured the ratio of public housing by dividing areas into those with a high and low proportion of public housing units relative to all households. The regional economic level was measured by categorizing areas into high and low economic levels, using the housing price index provided by the Seoul Metropolitan Government.

## Empirical model

A binary logistic regression model was used as analysis methodology in which a dependent variable answered 'yes' or 'no' to stigmatization and applied a robustness test. For the autonomous districts $j$ to which each respondent $i$ belongs, the formula for experiences of discrimination while living in public housing is provided below. $Pr(Y_{ijt})$ represents the probability that the respondent's experience of discrimination is 'yes,' serving as the dependent variable for individual respondent $i$ in autonomous district $j$. $Public\_Housing_{ij}$ is a dummy variable for types of public housing, based on SSPH, with IPH and SMPH as the reference categories. $Social\_Contact_{ij}$ encompasses variables related to an individual's perception of their relationship level with neighbors. $Regional\_Context_j$ includes variables for the regional economic level and the scale of public housing, while $X_{ij}$ accounts for the individual's socioeconomic characteristics. For the first research hypothesis, the first empirical analysis investigates the impact of SSPH strategies on experiences of stigmatization. This part explores the statistical differences in the experience of discrimination between those living in SSPH and those in non-SSPH settings. For this analysis, $Public\_Housing_{ij}$ is a dummy variable for SSPH versus non-SSPH, where non-SSPH is a composite variable that includes both IPH and SMPH. In addition to the

public housing variable, the model incorporates variables for social contact, regional context, and personal characteristics (Eq 1). In the second research hypothesis section, the next empirical analysis further dissects the non-SSPH category of $Public\_Housing_{ij}$ into IPH and SMPH. It is necessary to analyze the non-SSPH category by subdividing it into IPH and SMPH, as SMPH also has benefits in reducing stigmatization as a type of public housing. Accordingly, the analysis was conducted in models comparing SSPH with IPH, and SSPH with SMPH.

$$log \frac{Pr(Y_{ij})}{1 - Pr(Y_{ij})}$$
$$= \beta_0 + \beta_1 Public\_Housing_{ij} + \beta_2 Social\_Contact_{ij} + \beta_3 Regional\_Context_j + \beta_4 X_{ij} + e_{ij} \quad (1)$$

For the third research hypothesis, as indicated in Eq 2, the analysis incorporated interaction terms between the $Public\_Housing_{ij}$ variable and the $Social\_Contact_{ij}$ variables. This analysis examines the statistical differences in experiences of stigmatization among SSPH residents compared to IPH and SMPH, based on the social contact variables. The statistical significance and the direction of the coefficients of the interaction terms, $Public\_Housing_{ij} * Social\_Contact_{ij}$, were assessed to determine the statistical differences in stigmatization experiences among SSPH residents according to their social contact variables. For example, if the interaction term is statistically significant and positive, it indicates that higher levels of social contact are associated with increased experiences of stigmatization among SSPH residents. Conversely, a negative relationship implies that higher social contact leads to reduced experiences of stigmatization for SSPH residents. The data and Stata code used for these analyses can be accessed via the following link: https://figshare.com/account/articles/26962807.

$$log \frac{Pr(Y_{ij})}{1 - Pr(Y_{ij})} = \beta_0 + \beta_1 Public\_Housing_{ij} + \beta_2 Social\_Contact_{ij} + \beta_3 Regional\_Context_j + \beta_4 X_{ij}$$
$$+ \beta_5 Public\_Housing_{ij} * Social\_Contact_{ij} + e_{ij} \quad (2)$$

## Results

### The effect of SSPH on stigmatization experiences

Table 3 presents the experiences of stigmatization among public rental housing residents, focusing on SSPH. Model 1 uses stigmatization experience as the dependent variable and includes the SSPH residence variable as an independent variable, while Model 2 incorporates all control variables. The results show that the SSPH residence variable is statistically significant not only in Model 1 but also in Model 2, which includes all control variables. The SSPH variable has a statistically significant negative relationship with stigmatization experiences, indicating that SSPH residents experience less stigmatization compared to residents of other public rental housing. This finding addresses the first research hypothesis of the study, which asks whether SSPH residence offers any benefits in terms of reduced stigmatization. The analysis confirms that SSPH residents indeed experience less stigmatization, a fact that is statistically significant (Research hypothesis 1).

### The effects of SSPH on stigmatization experience comparisons of IPH and SMPH and its characteristics based on social contact variables

The next analysis addresses Research Hypothesis 2, empirically testing whether the benefits of reduced stigmatization experiences associated with SSPH residency are evident not only in

**Table 3. Analysis results on stigmatization experiences for the SSPH residents.**

| Variables | Model 1 | | | Model 2 | | |
|---|---|---|---|---|---|---|
| | Coef. | R.S.E. | P>z | Coef. | R.S.E. | P>z |
| Public housing | | | | | | |
| Scatter site public housing (ref. Other public housing) | -1.078*** | 0.238 | 0.000 | -1.233*** | 0.285 | 0.000 |
| Sociodemographic attributes | | | | | | |
| Age | | | | -0.037*** | 0.008 | 0.000 |
| Gender (male = 1, female = 0) | | | | -0.762*** | 0.186 | 0.000 |
| Education | | | | 0.327 | 0.220 | 0.138 |
| Employed | | | | -0.289 | 0.377 | 0.445 |
| Self-rate physical health | | | | -0.115 | 0.099 | 0.244 |
| Self-rate mental health | | | | -0.397*** | 0.144 | 0.006 |
| Household condition attributes | | | | | | |
| Log(household income) | | | | -0.147 | 0.147 | 0.317 |
| Married | | | | 0.234 | 0.220 | 0.289 |
| Number of family members | | | | -0.034 | 0.095 | 0.722 |
| Disable person | | | | 0.028 | 0.251 | 0.912 |
| Living period (year) | | | | 0.017 | 0.012 | 0.159 |
| Social mix and relationship attributes | | | | | | |
| Participate in communities | | | | 0.794** | 0.390 | 0.041 |
| Number of close neighbours | | | | -0.103 | 0.164 | 0.531 |
| Relationship degree with neighbours | | | | -0.028 | 0.144 | 0.846 |
| Regional context attributes | | | | | | |
| Public housing ratio | | | | 0.115 | 0.175 | 0.511 |
| Regional housing price index | | | | 0.146 | 0.176 | 0.407 |
| N | 700 | | | 700 | | |
| Log-likelihood | -443.089 | | | -401.962 | | |
| Pro > Chi$^2$ | <0.000 | | | <0.000 | | |
| Pseudo R$^2$ | 0.026 | | | 0.117 | | |

Note. Coef. = coefficient, R.S.E. = Robust standar error.

***P<0.01; **P<0.05; *P<0.1

SSPH = Scatter-site public housing

comparison with IPH but also with SMPH. Additionally, for Research Hypothesis 3, the study examines whether social contact helps mitigate stigmatization experiences among SSPH residents. Table 4 presents the stigmatization experiences of SSPH and IPH residents, while Table 5 displays the experiences of SSPH and SMPH residents. In each table, Model 1 uses stigmatization experience as the dependent variable and only includes the SSPH residency status as an independent variable, whereas Model 2 incorporates all control variables. Model 3 analyzes the characteristics of stigmatization according to social contact among SSPH residents by including interaction terms between SSPH residency status and social contact variables in the model.

The analysis results showed that the SSPH residence variable was statistically significant in Table 4, which compares it with IPH, as well as in Table 5, which compares it with SMPH. The SSPH variable is negatively correlated with stigmatization experiences, indicating that SSPH residents experience less stigmatization compared to both IPH and SMPH residents. This addresses the second research hypothesis of the study, which explores whether the benefits of

**Table 4. Analysis results on stigmatization experiences for the SSPH residents comparisons of IPH.**

| Variables | Model 1 | | | Model 2 | | | Model 3 | | |
|---|---|---|---|---|---|---|---|---|---|
| | Coef. | R.S.E. | P>z | Coef. | R.S.E. | P>z | Coef. | R.S.E. | P>z |
| Public housing | | | | | | | | | |
| Scatter site public housing (S) (ref. Independent public housing) | -1.089*** | 0.249 | 0.000 | -1.433*** | 0.343 | 0.000 | -6.202*** | 2.014 | 0.002 |
| Sociodemographic attributes | | | | | | | | | |
| Age | | | | -0.045*** | 0.010 | 0.000 | -0.045*** | 0.010 | 0.000 |
| Gender (male = 1, female = 0) | | | | -0.684*** | 0.239 | 0.004 | -0.753*** | 0.247 | 0.002 |
| Education | | | | 0.716** | 0.278 | 0.010 | 0.767*** | 0.282 | 0.007 |
| Employed | | | | -0.347 | 0.513 | 0.499 | -0.351 | 0.548 | 0.522 |
| Self-rate physical health | | | | -0.197 | 0.124 | 0.111 | -0.178 | 0.124 | 0.152 |
| Self-rate mental health | | | | -0.420** | 0.199 | 0.035 | -0.409** | 0.203 | 0.044 |
| Household condition attributes | | | | | | | | | |
| Log(household income) | | | | -0.235 | 0.190 | 0.216 | -0.235 | 0.193 | 0.223 |
| Married | | | | 0.183 | 0.290 | 0.528 | 0.242 | 0.293 | 0.409 |
| Number of family members | | | | -0.047 | 0.121 | 0.695 | -0.065 | 0.123 | 0.596 |
| Disable person | | | | 0.155 | 0.306 | 0.614 | 0.242 | 0.312 | 0.439 |
| Living period (year) | | | | 0.017 | 0.016 | 0.291 | 0.022 | 0.017 | 0.196 |
| Social contact attributes | | | | | | | | | |
| Participate in communities (P) | | | | 0.208 | 0.600 | 0.729 | 0.176 | 0.663 | 0.791 |
| Number of close neighbours (S) | | | | -0.180 | 0.217 | 0.406 | -0.130 | 0.242 | 0.591 |
| Relationship degree with neighbours (R) | | | | 0.400** | 0.187 | 0.032 | 0.161 | 0.219 | 0.463 |
| Regional context attributes | | | | | | | | | |
| Public housing ratio | | | | -0.190 | 0.246 | 0.441 | -0.198 | 0.246 | 0.421 |
| Regional housing price index | | | | 0.064 | 0.244 | 0.794 | 0.087 | 0.245 | 0.722 |
| Interaction term | | | | | | | | | |
| S*P | | | | | | | 0.340 | 1.319 | 0.797 |
| S*N | | | | | | | -0.047 | 0.623 | 0.940 |
| S*R | | | | | | | 1.716** | 0.744 | 0.021 |
| N | 458 | | | 458 | | | 458 | | |
| Log-likelihood | -280.967 | | | -244.104 | | | -239.959 | | |
| Pro > Chi² | <0.000 | | | <0.000 | | | <0.000 | | |
| Pseudo R² | 0.037 | | | 0.163 | | | 0.178 | | |

Note. Coef. = coefficient, R.S.E. = Robust standar error. ***P<0.01; **P<0.05; *P<0.1

IPH = Independent public housing, SSPH = Scatter-site public housing

SSPH in reducing stigmatization are evident not only compared to IPH but also to SMPH. The findings confirm that the reduction in stigmatization experienced by SSPH residents is statistically significant compared to both IPH and SMPH residents (Research hypothesis 2).

The results regarding the stigmatization characteristics associated with social contact among SSPH residents are as follows. The analysis showed that there were statistically significant differences in the characteristics of social contact among SSPH residents in the 'Relationship degree with neighbor' variable. Both Tables 4 and 5 indicated that the interaction between the SSPH variable and the 'Relationship degree with neighbor' variable was statistically significant and positive. According to the interaction term graphs, as shown in Figs 1 and 2, SSPH residents experience increased stigmatization as the 'Relationship degree with neighbor' increases. This addresses the third research hypothesis of the study, which explores whether

**Table 5. Analysis results on stigmatization experiences for the SSPH residents comparisons of SMPH.**

| Variables | Model 1 | | | Model 2 | | | Model 3 | | |
|---|---|---|---|---|---|---|---|---|---|
| | Coef. | R.S.E. | P>z | Coef. | R.S.E. | P>z | Coef. | R.S.E. | P>z |
| Public housing | | | | | | | | | |
| Scatter site public housing (S) (ref. Social mixed public housing) | -1.063*** | 0.258 | 0.000 | -0.984*** | 0.343 | 0.004 | -7.323** | 3.240 | 0.024 |
| Sociodemographic attributes | | | | | | | | | |
| Age | | | | -0.024** | 0.010 | 0.020 | -0.025** | 0.011 | 0.022 |
| Gender (male = 1, female = 0) | | | | -0.843*** | 0.261 | 0.001 | -0.918*** | 0.273 | 0.001 |
| Education | | | | -0.203 | 0.313 | 0.516 | -0.188 | 0.326 | 0.564 |
| Employed | | | | 0.055 | 0.552 | 0.920 | 0.086 | 0.567 | 0.880 |
| Self-rate physical health | | | | 0.042 | 0.144 | 0.772 | 0.052 | 0.145 | 0.722 |
| Self-rate mental health | | | | -0.281 | 0.195 | 0.150 | -0.304 | 0.205 | 0.138 |
| Household condition attributes | | | | | | | | | |
| Log(household income) | | | | -0.004 | 0.212 | 0.986 | -0.003 | 0.217 | 0.989 |
| Married | | | | 0.056 | 0.312 | 0.858 | 0.245 | 0.327 | 0.454 |
| Number of family members | | | | 0.123 | 0.155 | 0.428 | 0.080 | 0.158 | 0.610 |
| Disable person | | | | -0.183 | 0.419 | 0.663 | -0.114 | 0.434 | 0.792 |
| Living period (year) | | | | 0.033 | 0.023 | 0.157 | 0.035 | 0.024 | 0.145 |
| Social contact attributes | | | | | | | | | |
| Participate in communities (P) | | | | 0.913* | 0.482 | 0.058 | 1.045** | 0.524 | 0.046 |
| Number of close neighbours (S) | | | | -0.085 | 0.240 | 0.724 | 0.051 | 0.273 | 0.852 |
| Relationship degree with neighbours (R) | | | | -0.225 | 0.199 | 0.259 | -0.629** | 0.250 | 0.012 |
| Regional context attributes | | | | | | | | | |
| Public housing ratio | | | | 0.323 | 0.261 | 0.216 | 0.293 | 0.268 | 0.275 |
| Regional housing price index | | | | 0.150 | 0.275 | 0.587 | 0.239 | 0.282 | 0.397 |
| Interaction term | | | | | | | | | |
| S*P | | | | | | | -0.449 | 1.188 | 0.705 |
| S*N | | | | | | | -0.399 | 0.613 | 0.515 |
| S*R | | | | | | | 2.325*** | 0.773 | 0.003 |
| N | 379 | | | 379 | | | 379 | | |
| Log-likelihood | -227.205 | | | -207.760 | | | -200.095 | | |
| Pro > Chi$^2$ | <0.000 | | | <0.000 | | | <0.000 | | |
| Pseudo R$^2$ | 0.040 | | | 0.122 | | | 0.154 | | |

Note. Coef. = coefficient, R.S.E. = Robust standar error. ***P<0.01; **P<0.05; *P<0.1

SMPH = Social mixed public housing, SSPH = Scatter-site public housing

social contact helps reduce stigmatization experiences among SSPH residents. The findings suggest that a high 'Relationship degree with neighbor' may be detrimental to stigmatization experiences for SSPH residents (Research hypothesis 3).

## Discussion and conclusion

This study explored the effect of the SSPH strategy on the stigma experienced by the residents of public rental housing. Specifically, this study empirically analyzed the stigmatization experiences among SSPH residents in comparison with those of IPH and SMPH residents, focusing on public rental housing in Seoul. Based on social contact theory, we additionally explored the stigma experience characteristics of SSPH residents according to the social interactions with the neighbours. Finally, this study examined the regional context's influence on the features of

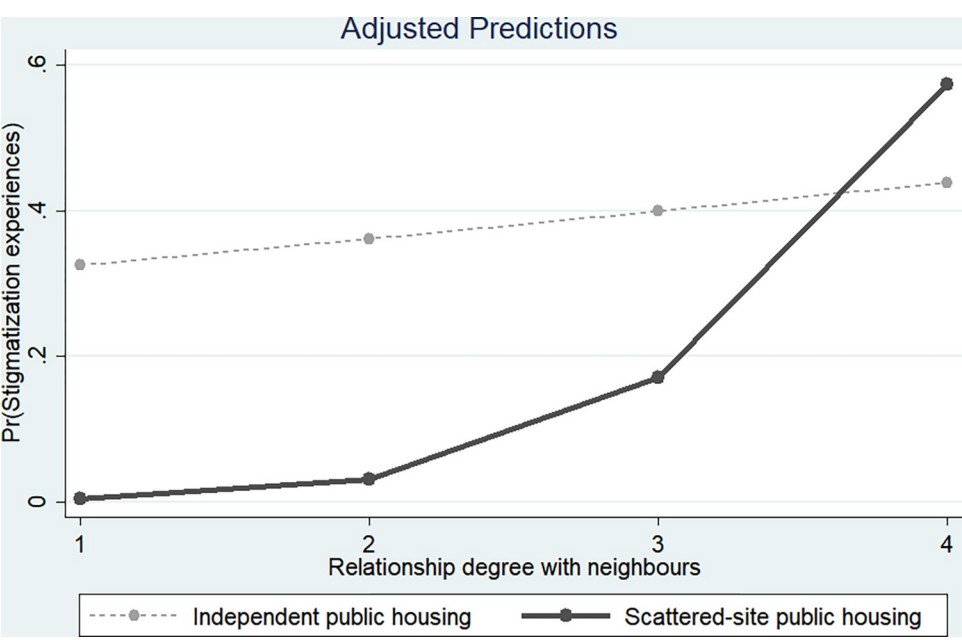

**Fig 1. Differences in stigmatization between SSPH and IPH according to relationship degree with neighbors.**

stigmatization experienced by residents living in public rental housing. The major findings of the study are as follows: It has been shown that SSPH residents experience statistically lower levels of stigmatization compared to residents of other public rental housing. Even residents of SMPH, which are provided specifically to reduce stigmatization, have higher experiences of stigmatization than SSPH residents, a fact that is also statistically significant. Additionally, in terms of social contact characteristics, SSPH residents experience increased stigmatization as

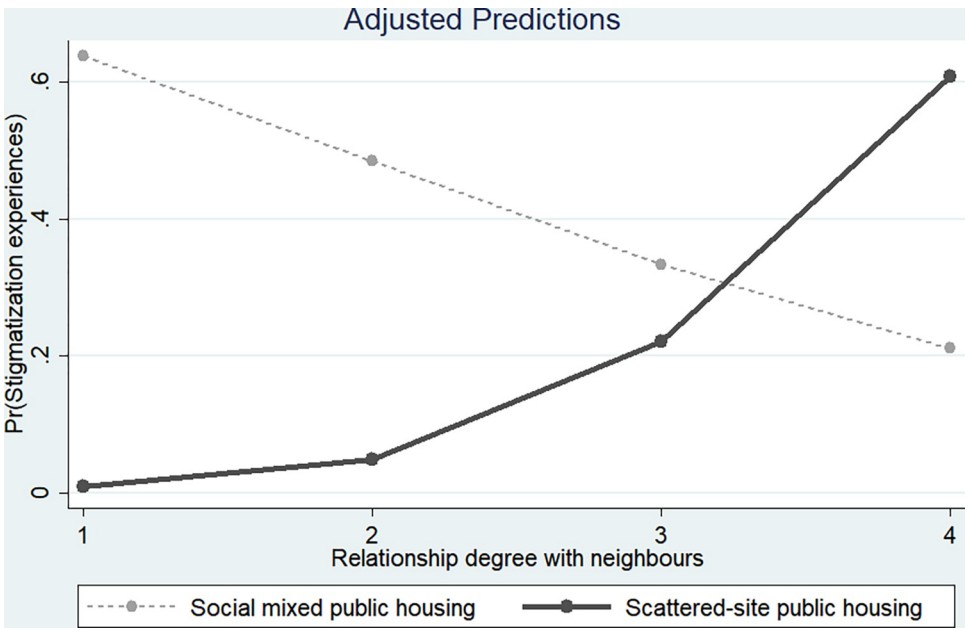

**Fig 2. Differences in stigmatization between SSPH and SMPH according to relationship degree with neighbors.**

their relationship degree with neighbors increases, compared to other public housing residents.

The results of this study indicate that SSPH positively affects stigma reduction. According to the first empirical analysis, SSPH residents experienced less stigmatization compared to residents of other public rental housing. This study quantitatively verifies the qualitative research arguments regarding SSPH residents' stigmatization [16, 17]. Furthermore, it statistically demonstrates that the SSPH strategy can effectively address discrimination, stigma, and social exclusion. While the stigma associated with public rental housing is often caused by the visual distinction between rental and non-rental housing [3, 4], an advantage of SSPH is that neighborhoods cannot visually identify whether homes are rented or privately owned [57]. Consequently, SSPH residents are statistically more likely to experience less stigmatization than residents of other rental housing. It is assumed that because the SSPH strategy removes visual distinctions, SSPH residents experience fewer stigmatization events. This finding illustrates that the SSPH system, which obfuscates whether a neighbor resides in public rental housing, effectively reduces the stigma experienced by SSPH residents.

In addition, SSPH residents experienced less stigmatization compared to IPH and SMPH residents. IPH residents could not shed the 'rental housing resident' label, as their district prominently displays the public rental housing brand 'Humansia'. Consequently, they face the stigma threat from residents of other general housing outside the district [23]. Although SMPH residents encounter reduced stigmatization from outside their district, they distinctly face new stigmatization from market-rate residents within the same district [1, 22]. In South Korea, IPH residents deal with negative behavior and shame associated with the 'Humansia' label, marking them as public rental housing residents [43]. Despite the SMPH residential district not being labeled 'Humansia', their residents still endure stigma, primarily from within their own community of public rental housing residents [58]. Therefore, the lesser experience of stigma among SSPH residents can be attributed to the absence of identifiable labels or indicators of public rental housing, factors that significantly contribute to reducing stigma compared to other public housing types.

The third research hypothesis investigates whether social contact variables positively impact the stigmatization experiences of SSPH residents. The analysis revealed that as relationships with neighbors deepen, stigmatization experiences tend to increase. This is particularly relevant in SSPH settings, where residents can more effectively conceal their public housing status unless they choose to disclose it. Such disclosures, when frequent interactions occur, can lead to increased stigmatization as residents' public housing status becomes known. According to the SPHD questionnaire, statistical results showed that residents who frequently interacted with their neighbors experienced more stigma than those who limited interactions to simple greetings. This finding aligns with Blokland's [59] emphasis on the stigma fear associated with revealing one's status as a rental housing resident. The study highlights that public rental housing residents often conceal their status to avoid stigma and are hesitant to form close relationships with neighbors. Excessive contact between public rental housing residents and residents of general housing can lead to negative outcomes [2, 35], a phenomenon that might also occur among SSPH residents. These results suggest that while developing social networks may aim to reduce stigma, it could paradoxically increase it by exposing residents' identities.

This study explored the SSPH strategy's effectiveness in reducing stigma in public rental housing, however it encompasses several limitations. Firstly, it mainly focuses on the perspectives of policymakers to evaluate the effectiveness of the SSPH strategy, rather than incorporating the views of residents. While the SSPH strategy's success in reducing stigmatization is evident, more in-depth research, including resident interviews, is necessary to determine its intrinsic benefits for those living in such housing. Put differently, reduced stigmatization does

not necessarily imply that low-income residents are fully satisfied living in SSPHs, or that the SSPH strategy is universally appropriate. While the SSPH approach may mitigate experiences of stigma, it serves more as a stigma-avoiding strategy than as a stigma-reducing one. Consequently, this study doesn't fully address the socio-ethical dimensions and residents' perceptions associated with these SSPH strategies. The second limitation of this study lies in the scope of its analysis on stigmatization, which was restricted to experiences of discrimination. Based on Sisson's [60] research, the occurrence of stigma related to public rental housing is very broad. Sisson underscores the importance of examining not just stigmatization itself, but also the mechanisms through which residents of public housing internalize stigma. These mechanisms can include influences from the political economy and media, which may contribute to creating a territorial stigma around public housing. However, this study did not extend its analysis to include these broader aspects of stigma. Finally, COVID-19 has caused major changes in community social relationships [61]. Since this study, based on the social context theory, used the SPHD data from 2019, that is, before the outbreak of COVID-19, we could not deal with the rapidly changing social relationships in recent years. In future, when the SPHD, reflecting on COVID-19, is published, it is necessary to explore the factors that contribute to the stigmatization of SSPH residents in the COVID-19 pandemic environment based on the social contact theory.

## Author Contributions

**Conceptualization:** Sungik Kang.

**Data curation:** Sungik Kang.

**Formal analysis:** Sungik Kang, Ja-Hoon Koo.

**Funding acquisition:** Sungik Kang.

**Methodology:** Sungik Kang.

**Project administration:** Sungik Kang.

**Supervision:** Ja-Hoon Koo.

**Validation:** Ja-Hoon Koo.

**Visualization:** Sungik Kang.

**Writing – original draft:** Sungik Kang.

**Writing – review & editing:** Sungik Kang, Ja-Hoon Koo.

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
