## [Decision Letter · Decision Letter 0]

1 Aug 2024

PONE-D-24-20986Exploring stigma experiences of scattered-site public housing residents and its characteristics based on social contact theoryPLOS ONE

Dear Dr. Koo,

Thank you for submitting your manuscript to PLOS ONE. After careful consideration, we feel that it has merit but does not fully meet PLOS ONE’s publication criteria as it currently stands. Therefore, we invite you to submit a revised version of the manuscript that addresses the points raised during the review process.

We look forward to receiving your revised manuscript.

Kind regards,

Kasi Eswarappa

Academic Editor

PLOS ONE

 [This work was supported by the Korea National Research Foundation under Grant [RS-2023-00239319].].  

3. Thank you for uploading your study's underlying data set. Unfortunately, the repository you have noted in your Data Availability statement does not qualify as an acceptable data repository according to PLOS's standards.

4. Please remove your figures from within your manuscript file, leaving only the individual TIFF/EPS image files, uploaded separately. These will be automatically included in the reviewers’ PDF.

Additional Editor Comments:

We advise the author(s) to carry out the minor revisions of the manuscript.

Reviewers' comments:

Reviewer's Responses to Questions

**Comments to the Author**

1. Is the manuscript technically sound, and do the data support the conclusions?

Reviewer #1: Yes

2. Has the statistical analysis been performed appropriately and rigorously? 

Reviewer #1: Yes

3. Have the authors made all data underlying the findings in their manuscript fully available?

Reviewer #1: Yes

4. Is the manuscript presented in an intelligible fashion and written in standard English?

Reviewer #1: Yes

5. Review Comments to the Author

Reviewer #1: Abstract: should be written again in academic forms. Main aim, where study conducted, model used, period, and main findings.

Introduction: The author (s) should highlight the contribution of this work to the existing literature review in the last part of the introduction. The written contribution is not clear and the statements are too long.

I propose in the introduction should specify the methodology of research and research hypotheses. The diagnosis itself should indicate the novelty of the results and to publish the considerations in scientific journals. It should define the purpose of the work and its significance, including specific hypotheses being tested. The current state of the research field should be reviewed carefully and key publications cited.

I suggest that you refrain from describing in the introduction what the author has included in the various parts of the article

Literature review: This research gap must be created by a systematic literature review that provides 'holes' in the state of knowledge on the topic. At the end of your justification you should write something like: According to what we have been able to find, there are no studies referring to and reporting on .... By doing so, you have demonstrated the relevance of the issue and also proved that your study does indeed fill a research gap.

6. PLOS authors have the option to publish the peer review history of their article (what does this mean?). If published, this will include your full peer review and any attached files.

Reviewer #1: No

---

## [Author Response · Author response to Decision Letter 0]

13 Oct 2024

1. Response

- We have reviewed the Title Page and ensured it follows the journal's guidelines.

- The main text has also been revised according to the guidelines for titles, figures, tables, and references.

 [This work was supported by the Korea National Research Foundation under Grant [RS-2023-00239319].]. 

2. Response

- We have prepared the Role of Funder statement as follows and included it in the cover letter.

- The funder had a role in study design, data collection, empirical analysis, and drafting the manuscript.

3. Thank you for uploading your study's underlying data set. Unfortunately, the repository you have noted in your Data Availability statement does not qualify as an acceptable data repository according to PLOS's standards.

At this time, please upload the minimal data set necessary to replicate your study's findings to a stable, public repository (such as figshare or Dryad) and provide us with the relevant URLs, DOIs, or accession numbers that may be used to access these data. For a list of recommended repositories and additional information on PLOS standards for data deposition.

3. Response

- We have uploaded the mentioned data set to Figshare. The link is as follows: https://figshare.com/account/articles/26962807

- Additionally, we have included the following statement in the manuscript.

- The data and Stata code used for these analyses can be accessed via the following link: https://figshare.com/account/articles/26962807.

4. Please remove your figures from within your manuscript file, leaving only the individual TIFF/EPS image files, uploaded separately. These will be automatically included in the reviewers’ PDF.

4. Response

- As you suggested, we have removed the figures from the manuscript file and will upload the image files separately.

5. Response

- As you pointed out, there were three issues related to reference citations, which we have corrected as follows:

- In the sentence "which is one of the biggest obstacles in operating public rental housing [2-4]," we changed the citation from [3-4] to [2-4].

- We swapped the order of references 58 and 59 in both the main text and the reference list.

- We removed reference 60 since it was not cited in the text, and adjusted the numbering: 61 was changed to 60, 62 to 61, and 63 to 62.

Review Comments to the Author

1. Comment

Reviewer #1: Abstract: should be written again in academic forms. Main aim, where study conducted, model used, period, and main findings.

1. Response: The reviewer suggested that the abstract include the research objective, study area, analytical model, key findings, and the year of the study data. While the study area, analytical model, and key findings were already well organized, the research objective was unclear, and the year of the study data was not specified. Therefore, we have clarified the research objective and added the year of the data used.

2. Comment 

Introduction: The author (s) should highlight the contribution of this work to the existing literature review in the last part of the introduction. The written contribution is not clear and the statements are too long.

I propose in the introduction should specify the methodology of research and research hypotheses. The diagnosis itself should indicate the novelty of the results and to publish the considerations in scientific journals. It should define the purpose of the work and its significance, including specific hypotheses being tested. The current state of the research field should be reviewed carefully and key publications cited.

I suggest that you refrain from describing in the introduction what the author has included in the various parts of the article

1) Comment: The contribution of this study to the existing literature should be emphasized. 

1) Response:

- The contribution of this study is described at the end of Chapter 2. We have revised the sentence to make it more emphasized and clearer. Additionally, the contribution of this study is already demonstrated as addressing areas not covered by previous research, including studies 16, 17, and 30, which are particularly similar to this study. Below is the final revised content related to this matter.

- Although various studies have examined the effects of SSPH on resident satisfaction [16], social interaction, and socio-economic status [30], as well as changes in home prices [17], there remains a significant gap in the literature concerning SSPH's role in reducing stigmatization. … …. Thus, this study contributes to the existing body of literature by providing a quantitative analysis of whether SSPH offers advantages in reducing stigmatization compared to other types of public housing such as IPH and SMPH. Additionally, this research sheds light on the influence of neighborhood relationships among SSPH residents on stigmatization, an area that has not been extensively explored in previous studies.

2) Comment: The research hypotheses should be specified.

2) Response:

- It appears that the reviewer believes it is more appropriate to state research hypotheses rather than research questions in this study.

- Originally, there were research questions, but since research questions and hypotheses of a similar nature do not need to be presented together, we have transformed the original research questions into hypotheses as follows.

- The hypotheses of this study are as follows. The first hypothesis is that SSPH actually shows advantages in reducing stigmatization. This analysis distinguishes between IPH and SMPH within public rental housing, as SMPH was introduced as a policy to mitigate stigmatization by mixing higher-income and public rental households within one residential community. However, despite reducing external stigmatization outside the residential district, SMPH also experiences internal stigmatization among residents [1]. Therefore, the second hypothesis is that SSPH shows advantages in reducing stigmatization not only in IPH but also in SMPH. The third hypothesis is that social contact conditions help SSPH residents reduce their experience of stigmatization. According to social contact theory and empirical research, social contact can have positive effects, but excessive contact may also have negative impacts. Hence, the third hypothesis examines the characteristics of stigmatization among SMPH residents based on social contact.

3) Comment: It was suggested to describe the novelty of the diagnosis.

3) Response:

- We have highlighted the novelty of the diagnosis by adding the following statement to the manuscript: "The diagnosis of these hypotheses is expected to reveal findings that were not empirically demonstrated in previous studies and may lead to unexpected conclusions."

4) Comment: It was suggested to specify the methodology of the study and to reduce the length of the paragraphs by removing the descriptions of the content included in each section of the paper.

4) Response:

- We have specified the study's methodology in the introduction as follows. Additionally, we have shortened the length of the paragraphs by removing unnecessary descriptions of each section's content.

- To conduct this empirical analysis, the methodology involves first addressing data imbalance and then using binary logistic regression to analyze and compare the levels of stigmatization in SSPH with those in IPH and SMPH.

5) Comment: The research objective and its significance should be clearly defined.

5) Response:

- We have revised the relevant section to clearly reflect the research objective, as shown below.

- Therefore, the purpose of this study is to explore whether residents of scattered-site public housing experience less stigmatization compared to residents of other public housing types.

- Specifically, it is significant to analyze the characteristics of stigmatization by comparing the levels experienced by SSPH residents with those experienced by residents in independent public housing (IPH), where only public housing residents live, and socially mixed public housing (SMPH).

6) Comment: Review and cite the current state of research in the field.

6) Response:

- In Chapter 2, we have reviewed all the significant studies related to this topic and logically presented the necessary content.

- Therefore, no additional actions were taken regarding this point.

3. Comment

Literature review: This research gap must be created by a systematic literature review that provides 'holes' in the state of knowledge on the topic. At the end of your justification you should write something like: According to what we have been able to find, there are no studies referring to and reporting on .... By doing so, you have demonstrated the relevance of the issue and also proved that your study does indeed fill a research gap.

3. Response: We have made the following revisions according to the reviewer's comments.

To our knowledge, no existing studies have statistically compared SSPH with other types of public housing, such as IPH and SMPH, to verify its effectiveness in this regard. By doing so, this study aims to fill this gap by analyzing the characteristics of SSPH related to stigmatization and demonstrating its potential advantages over other housing types.

---

## [Editor Report · Decision Letter 1]

17 Oct 2024

Exploring stigma experiences of scattered-site public housing residents and its characteristics based on social contact theory

PONE-D-24-20986R1

Dear Dr. Koo,

We’re pleased to inform you that your manuscript has been judged scientifically suitable for publication and will be formally accepted for publication once it meets all outstanding technical requirements.

Kind regards,

Kasi Eswarappa

Academic Editor

PLOS ONE

Additional Editor Comments (optional):

I appreciate your effort in revising the paper. Hence, I am accepting it for publication in our respected journal.
---

## [Editor Report · Acceptance letter]

22 Oct 2024

PONE-D-24-20986R1 

PLOS ONE

Dear Dr. Koo, 

I'm pleased to inform you that your manuscript has been deemed suitable for publication in PLOS ONE. Congratulations! Your manuscript is now being handed over to our production team.

Kind regards, 

on behalf of

Dr. Kasi Eswarappa 

Academic Editor

PLOS ONE